# Knowledge and attitudes toward complete diagnostic autopsy and minimally invasive autopsy: A cross-sectional survey in Hanoi, Vietnam

**Ngan Ta Thi Dieu**[1,2‡], **Nhung Doan Phuong**[3‡], **My Nguyen Le Thao**[3],
**Mary Chambers**[3,4], **Duy Manh Nguyen**[5], **Ha Thi Lien Nguyen**[1,2], **Huong Thi Thu Vu**[1],
**Thach Ngoc Pham**[1], **Rogier van Doorn**[4,6¶], **Jennifer Ilo Van Nuil**[3,4¶]*

**1** National Hospital for Tropical Diseases, Ha Noi, Vietnam, **2** Department of Infectious Diseases–Hanoi Medical University, Ha Noi, Vietnam, **3** Oxford University Clinical Research Unit, HCMC, Vietnam, **4** Nuffield Department of Medicine, University of Oxford, Oxford, United Kingdom, **5** Grinnell College, Grinnell, Iowa, United States of America, **6** Oxford University Clinical Research Unit, Ha Noi, Vietnam

‡ NTTD and NDP are joint first authors on this work.
¶ RD and JIN are joint last authors on this work.
* jvannuil@oucru.org

**Data Availability Statement:** The data for this analysis are included as supporting information at the end of the manuscript.

## Abstract

Knowing the cause of death (CoD) plays an important role in developing strategies and interventions to prevent early mortality. In Vietnam, the CoD of the majority of patients who acquired infectious diseases remains unknown. While there are challenges that hinder the use of complete diagnostic autopsy (CDA) in practice, minimally invasive autopsy (MIA) might be a promising alternative to establish CoD in Vietnam. The current study aims to explore knowledge of and attitudes toward CDA and MIA in the wider population in Vietnam. The study was cross-sectional, using structured questionnaires that were disseminated electronically via several websites and as paper-based forms in a national level hospital in Vietnam. Descriptive analyses were performed and where appropriate, comparisons between the healthcare workers and the general public were performed. We included 394 questionnaires in the analysis. The majority of participants were under age 40, living in major cities and currently practicing no religion. 76.6% of respondents were aware of CDA and among them, 98% acknowledged its importance in medicine. However, most participants thought that CDA should only be performed when the CoD was suspicious or unconfirmed because of its the invasive nature. For MIA, only 22% were aware of the method and there was no difference in knowledge of MIA between healthcare workers and the wider public. The questionnaire results showed that there are socio-cultural barriers that hinder the implementation of CDA in practice. While the awareness of MIA among participants was low, the minimally invasive nature of the method is promising for implementation in Vietnam. A qualitative study is needed to further explore the ethical, socio-cultural and/or religious barriers that might hinder the implementation of MIA in Vietnam.

**Funding:** This research was funded by the Viet Nam Call in Infectious Diseases, MRC Newton Fund (UK) and Ministry of Science and Technology (VN): Establishing and evaluating minimally invasive autopsy to determine cause of death from infectious disease in Viet Nam. The work was supported by MRC (MR/R026300/1 to RvD) and MOST (NT.78.UK/20 to NTTD). The funders played no role in the design, data collection, analysis, or write-up of this research.

**Competing interests:** The authors have declared that no competing interests exist.

**Abbreviations:** CaDMIA, ; CDA, Complete diagnostic autopsy; CoD, Cause of death; HCMC, Ho Chi Minh City; HCW, Healthcare workers; LMICs, Low and middle-income countries; MIA, Minimally invasive autopsy; MITS, Minimally invasive tissue sampling; NHTD, National Hospital for Tropical Diseases; OxTREC, Oxford Tropical Research Ethics Committee; PERCH, Pneumonia Etiology Research for Child Health; VA, Verbal autopsy; WHO, World Health Organization.

## Introduction

Knowledge of the cause of death (CoD) is an important public health metric to guide interventions and research. Infectious diseases are estimated to be the cause of 15–20% of all mortality in Vietnam but the exact aetiology of the CoD in these cases remains unknown in the majority of patients (>50%) [1, 2]. Recent studies from Vietnam have also shown that, despite a wide arsenal of diagnostic assays used in these studies, an exact aetiology of major infectious syndromes cannot be established in a large proportion of patients (>50% in central nervous system infections and 30% in respiratory infections necessitating hospitalization) [3, 4].

In routine hospital practice, this incomplete diagnosis is due both to a lack of diagnostic capacity and difficulty obtaining the right specimens. For example, for encephalitis, a brain biopsy would be the optimal specimen; whereas in routine diagnostics clinicians rely on cerebrospinal fluid investigation and blood cultures, which often do not contain the causative agent. It is equally difficult to obtain representative samples from the lung parenchyma or deep respiratory tract to accurately diagnose the aetiology of pneumonia. Optimal samples are difficult to obtain both in terms of technical skills and an unacceptable risk:benefit ratio for individual patients. Research determining causes of pneumonia and encephalitis in the autopsy setting may help determine common aetiological agents in a particular community, guiding therapy in living patients and informing public health policy measures such as vaccination programmes [5].

Complete diagnostic autopsy (CDA) is considered the gold standard to establish CoD. The method involves observations of both the outside and the inside of the body, alongside open dissection to acquire tissues from the organs [6]. Several relevant analyses would then be performed on the collected samples, including toxicology, microbiology, genetics, etc [6]. Despite its benefits, this post-mortem procedure is rarely performed in Vietnam. The reasons for this are threefold: first, most deaths occur outside the health system (i.e. patients are often discharged to die at home when prognosis is unfavourable); second, there are no facilities or trained personnel for full autopsy and subsequent pathological and microbiological examination in most provincial and referral hospitals; and finally, a major factor limiting consent to autopsy are negative attitudes toward post-mortem examination, due to religious beliefs and cultural traditions (1). Verbal autopsy (VA), a structured interview of relatives and caretakers, is recommended by World Health Organization (WHO) as an alternative to CDA in low and middle-income (LMIC) countries. VA has been used in Vietnam [1]. VA provides a broad syndromic diagnosis and is concordant with physician coding, but its performance to establish a specific CoD is poor and the validity and adequacy of VA is under debate [6].

Minimally invasive autopsy (MIA) or minimally invasive tissue sampling (MITS), in which hollow needles are used to sample a selection of body fluids and organs, has been proposed as an alternative to CDA. Recently, large studies such as the "Validation of the minimally invasive autopsy tool for cause of death investigation in developing countries" (CaDMIA) [7]. The "Pneumonia Etiology Research for Child Health" (PERCH) trials [8] suggest that the results of MIA are comparable to CDA [9, 10]. Both studies showed high rates of success (>80%) of identifying a CoD using percutaneous needle biopsy followed by the use of the resultant samples for a range of diagnostic techniques, such as standard histopathology, culture and molecular microbiological techniques for aetiological diagnosis in infectious disease deaths, rather than open autopsy [9, 10].

A simplified, standardized MIA protocol, without the need for sophisticated imaging techniques such as computed tomography scan or magnetic resonance imaging, was recently described and validated. The sampling procedure which uses portable ultrasound and hollow needle necropsy from body fluids and major organs was shown to obtain adequate material

and to lead to accurate diagnoses [11]. A recent study compared MIA and CDA in Mozambique and showed high yield and concordance, especially among infectious diseases (80%) with the aetiology established in 84% in the MIA group [7].

Results from previous studies showed that the implementation of MIA in Vietnam might be promising. However, alongside the clinical component, it is essential to explore the context specific concepts of death and dying, as well as the rituals and norms that might affect the acceptability of the method in this context. In a study with Thai, Muong and Hmong participants in Vietnam, the authors argued that all three groups shared a similar "beliefs of life cycle" that does not consider death the ultimate endpoint [12]. Instead, they believe when a person passes away, the soul leaves the body but continues to live in another form in another world where they can still maintain close contact, such as supporting or harasing, with those who are living. The rituals, including the funeral, thus, would not be treated as the family's private business only but also an obligation for people outside the family or the village, to partcipate therefore proving its social nature [12]. In An Giang–a province located at Southwest region of Vietnam, members of Cham communities viewed death and organized death rituals according to their Islamic beliefs. For them, life was only a "fleeing realm" so death was not concerning because the deceased would be sent back to God [13]. These norms are common within Vietnamese traditional beliefs more broadly and therefore are important to consider the circumstances under which CDA or MIA could fit into longstanding beliefs [14].

The work described here represents the first step in a larger body of work in which we intend to establish and evaluate a procedure for MIA to establish CoD in Vietnam. To be able to deliver a consent procedure and a protocol for MIA in a Vietnamese context, we first conducted a baseline study exploring the specific Vietnamese cultural and religious factors that could impact consent and create barriers toward autopsy in general and MIA specifically. The baseline study included a survey and a qualitative component. In this article, we report the results from the survey, which was conducted as the first component of the baseline study to explore the wider public's perceptions about CDA and MIA.

## Materials and methods

### Ethics statement

The study was approved by Oxford Tropical Research Ethics Committee (OxTREC) in Oxford, UK and the ethics committee at the National Hospital for Tropical Diseases in Hanoi, Vietnam. All methods were performed in accordance with the relevant guidelines and regulations. By completing the questionnaire, the participants agreed to participate in the study, therefore, no additional signed informed consent forms were collected for this component.

We conducted a cross-sectional study to explore knowledge and attitudes of CDA and MIA. Structured questionnaires were developed based on a review of previous literature on the communities' perception of CDA and MIA in low- and middle-income countries (LMICs) and the socio-cultural context of Vietnam. We included questions about the participants' demographics, including age, occupation, city/province of residence, but no identifying information was collected. The questionnaire included Yes/No, single and multiple-choice questions and covered topics on the participants' awareness of and attitudes toward CDA and MIA, as well as barriers to CDA and MIA implementation in Vietnam. A brief description of CDA and MIA in lay language was included at the beginning of the survey.

Data collection was conducted over three months, from May to July 2021, via two methods: (1) a web-based questionnaire available to the general public advertised on the website of the National Hospital for Tropical Diseases (NHTD) and several online newspapers and (2) a paper-based questionnaire distributed to hospital staff and patients coming to the examination

rooms at NHTD, Hanoi. All adults who were 18 years old or above and agreed to answer the questionnaire were eligible to take part in the study. We excluded people who were under 18 years old and/or did not have mental capacity to complete the questionnaire.

We aimed for 500 questionnaires returned. The goal of the questionnaire was to obtain initial understanding of what people know and think about CDA and MIA, instead of examining a representative sample that reflects the awareness and attitudes of the whole population, no formal sample size calculation was done. Data analysis was conducted using IBM SPSS Statistics 25. Quantitative variables were reported as mean and standard deviation, whereas qualitative variables were illustrated as frequencies and percentages. Following a thorough literature review, characteristics such as age, residential location, educational level and religion might affect an individual's perception on death, dying and the body. These socio-demographic factors, therefore, were included to enable comparisons between groups later on. Participants' age was transformed into categorical variable with four groups: (1) Under 30; (2) from 30 to 39 years old; (3) from 40 to 49 years old and (4) from 50 years old and above. Residential location was grouped into two groups: (1) Hanoi/Ho Chi Minh City or (2) Other provinces. Occupation groups included (1) Healthcare workers (HCWs) and (2) Community. We also performed analysis to compare the awareness and attitudes regarding CDA and MIA between HCWs and community members (which included all other occupation groups). Chi-square tests or Fisher's exact tests were done appropriately to determine the difference between these two groups. In some specific questions regarding the attitudes towards CDA and MIA, additional descriptive analyses were performed to further investigate the different perceptions across the socio-demographic subgroups within the general public group.

## Results

### Socio-demographical characteristics of the participants

A total of 500 questionnaires were collected over the three-month period, of which 106 were excluded from the analysis due to incompleteness or selection of more than one response for a single question over multiple questions. 394 questionnaires were included in the analysis. Table 1 shows socio-demographic characteristics of the study population. We also aimed to explore the awareness of and attitudes towards the autopsy methods among HCWs and the general public separately, therefore the table also includes the characteristics of each group.

As an overview, the majority of respondents were under 40 years of age (63.2%, mean age of 37.09) and living in either Hanoi or Ho Chi Minh City (HCMC) (52.0%). Almost two-thirds (64.7%) of the participants had earned a university degree. The proportion of participants who did not practice any religion was 76.4%. For occupation, nearly 20% of the respondents were working in the healthcare sector. Within the HCW group, more than 80% of the participants were under 40 years of age. All of the HCWs had at least a university degree and most of them did not practice any religion. The community group, on the other hand, had higher proportions of older people and people who finished secondary school or high school education. The number of community members who were either Buddhist or Christian was also slightly higher than in the HCW group.

### Awareness and attitude in relation to complete diagnostic autopsy

**Awareness of CDA.**    302 out of 394 (76.6%) participants reported that they had heard of CDA as a method to determine CoD, while 23.3% had not. There was a difference between the HCW and the community groups in the awareness of CDA as a method to determine CoD (p<0.05), with 98.6% of the HCWs aware of CDA in comparison to 71.6% in the other group.

**Table 1. Socio-demographic characteristics of study participants.**

| Socio-demographic variables | | Healthcare workers (n₁ = 74) | Community (n₂ = 320) | Total (N = 394) |
|---|---|---|---|---|
| | | Mean (SD) | Mean (SD) | Mean (SD) |
| **Age** | | 33.6 (8.7) | 37.9 (12.7) | 37.09 (12.2) |
| | | n (%) | n (%) | n (%) |
| **Age groups** | Under 30 | 28 (40.0) | 92 (28.7) | 120 (30.5) |
| | 30–39 | 31 (41.8) | 98 (30.7) | 129 (32.7) |
| | 40–49 | 12 (16.2) | 80 (25.0) | 92 (23.3) |
| | 50 and above | 3 (4.0) | 50 (15.6) | 53 (13.5) |
| **City/Province of residence** | Hanoi/HCMC | 39 (52.7) | 166 (51.9) | 205 (52.0) |
| | Other cities/provinces | 35 (47.3) | 154 (48.1) | 189 (48.0) |
| **Education** | Secondary school | 0 (0.0) | 40 (12.5) | 40 (10.2) |
| | High school | 0 (0.0) | 99 (30.9) | 99 (25.1) |
| | Undergraduate/ Postgraduate | 74 (100.0) | 181 (66.6) | 255 (64.7) |
| **Religion** | None | 67 (90.5) | 234 (73.1) | 301 (76.4) |
| | Buddhism | 5 (6.8) | 71 (22.2) | 76 (19.3) |
| | Christianity | 2 (2.7) | 15 (4.7) | 17 (4.3) |

"Fig 1" displays the specific benefits of conducting CDA. In general, most of the participants recognized at least one benefit of CDA, leaving only 1.5% of the participants who saw no benefits in conducting CDA. 72.1% of participants stated that determining an accurate CoD of an individual was a benefit of CDA, followed by 51.5% of the participants listing detection of hereditary diseases at early stages for optimal treatment plans as a benefit. A slightly smaller

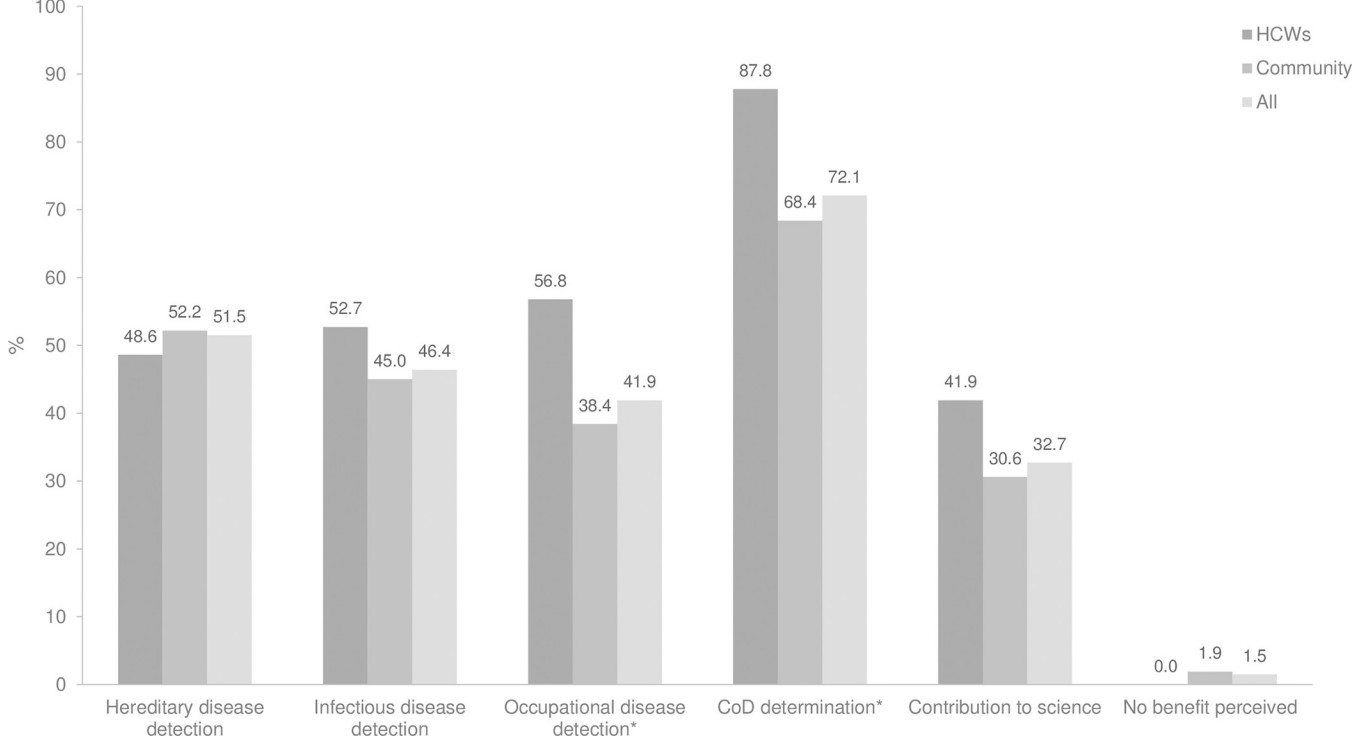

**Fig 1. Benefits of CDA, as perceived by the participants.**

proportion (46.5%) of participants thought that CDA could help infectious disease detection for contact tracing, disease control and treatment. However, only a third of the respondents (32.7%) thought that performing CDA could contribute to science and save lives. There were statistical differences in the awareness between the HCWs and the community in terms of the potential of CDA to detect occupational diseases or to investigate CoD. Although no significant difference was found, the result showing the different awareness regarding CDA's contribution to science between the HCWs and the community group was worth highlighting.

**Attitudes towards CDA.**   For the participants who had previously heard of CDA (n = 302), we asked questions about the perceived importance of CDA. Among these, almost all (98.7%) of the participants perceived the importance of CDA in forensic medicine specifically, while almost all (96.0%) perceived its importance in medicine more generally. There was no difference between the HCW and the community groups.

All participants were asked about the situations in which conducting a CDA was necessary. 281 participants (71.3%) thought that CDA should be conducted in situations related to deaths where the CoD was not clear, including deaths with (forensically) suspicious causes and deaths with unconfirmed diagnoses. Only around half of the respondents thought that CDA should be conducted when it was requested by either the family (51.5%) or the authorities (49.5%) ("Fig 2"). Across all the domains, the percentages in the HCW group were higher than in the community group. Particularly, while 75.7% of HCWs thought that a CDA should be performed when the authority requested it, only 43.4% of the community group agreed.

In relation to the reasons for objecting CDA, around three quarters (75.48%) of the participants stated that conducting CDA would cause an invasion (*xâm phạm thi thể*), for example, unnecessary cutting to the body of the deceased. There was no difference between the two groups regarding this belief. On the other hand, more than 50% said that performing CDA

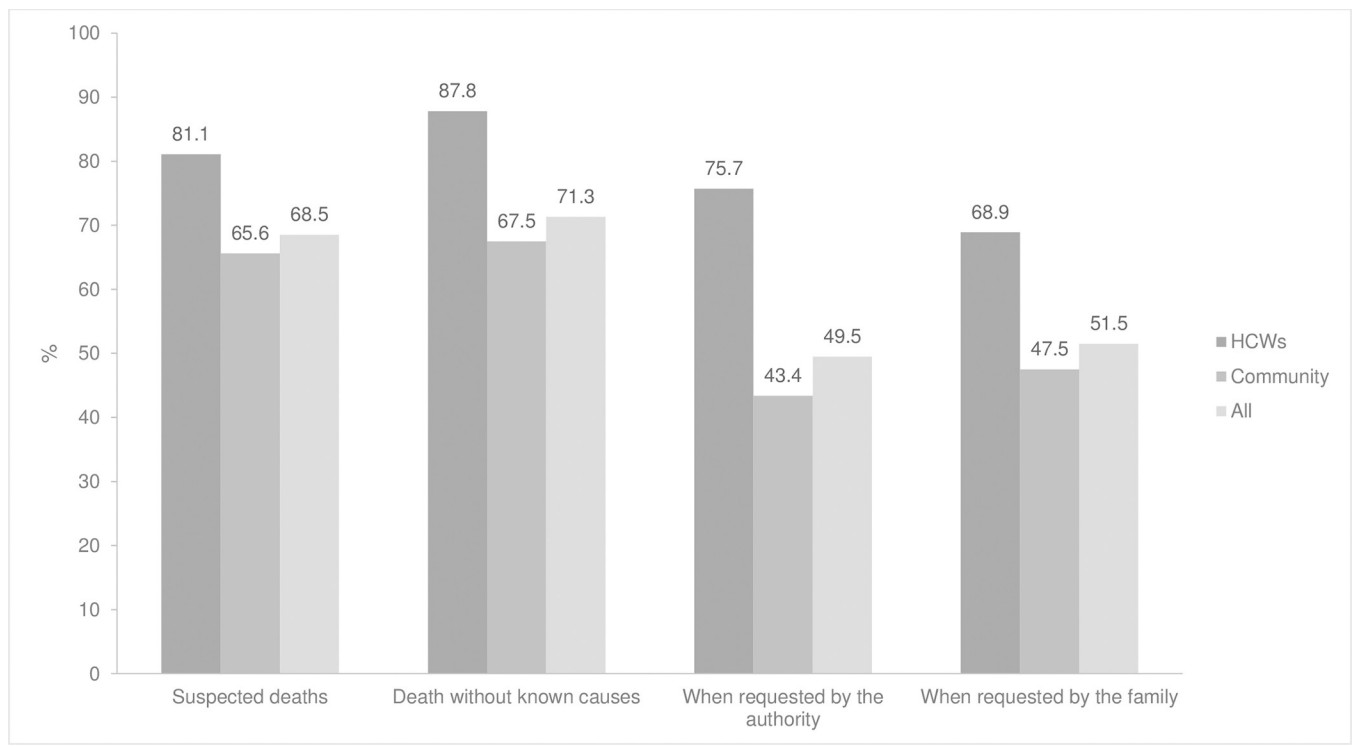

**Fig 2. Situations when CDA should be performed.**

**Table 2. Reasons for objecting CDA between HCWs and community members.**

| Reasons for objecting CDA | HCWs (n₁ = 74) | Community (n₂ = 320) | p | Total (N = 394) |
|---|---|---|---|---|
|  | n (%) | n (%) |  | n (%) |
| CDA is against religion and/or traditions and customs of the family | 56 (75.7) | 158 (49.4) | < **0.001** | 214 (54.3) |
| CDA causes invasion to the dead body | 61 (82.4) | 236 (73.8) | 0.14 | 297 (75.4) |
| CDA affects burial plans | 24 (32.4) | 57 (17.8) | **0.007** | 81 (20.6) |
| Clinical diagnoses are trusted | 4 (5.4) | 30 (9.4) | 0.36 | 34 (8.6) |
| CoD is not important | 12 (16.2) | 62 (19.4) | 0.62 | 74 (18.8) |

would be against their religious faith, traditions or customs. There were 74 respondents (18.8%) who answered that knowing CoD was not important, therefore, CDA should not be performed. There were differences between the HCW and the community groups for two concerns: (1) CDA could violate religion and/or traditions and customs of the family and (2) CDA affects burial plans of the family (Table 2).

The importance of CDA for medicine and forensic medicine was perceived by most of the respondents who had had previous awareness of the method, however, the majority of the respondents thought that CDA was only needed in cases where CoD was unclear. Reasons for the unfavorable attitudes of the participants towards the method were mostly due to its invasive nature and socio-cultural as well as religious beliefs. To explore further the different beliefs among socio-demographic groups, descriptions of perceptions across the community group are shown in Table 3.

Across the socio-demographic subgroups within the community group, the body disfiguring nature of CDA appeared as the main concern, followed by the potential violation of CDA to the family's religion, traditions or customs. The proportions of people under 40, living in either two major cities or having a university degree who had concerns about the socio-cultural barriers against CDA were higher than the other groups. On the other hand, the percentage of the people who attended higher education perceived that knowing the CoD was not important was more considerable than in the other two groups.

**Table 3. Reasons for opposing against CDA across socio-cultural groups among the community respondents (N = 320).**

| | | Religion/ Traditions/ Customs violation | Body invasion | Delay to burial plans | Trust in initial clinical diagnoses | Knowing CoD is unnecessary |
|---|---|---|---|---|---|---|
| | | n (%) | n (%) | n (%) | n (%) | n (%) |
| **Age group** | Under 30 | 55 (17.2) | 67 (20.9) | 24 (7.5) | 6 (1.9) | 26 (8.1) |
| | 30–39 | 54 (16.9) | 70 (21.9) | 19 (5.9) | 11 (3.4) | 18 (5.6) |
| | 40–49 | 31 (9.7) | 62 (19.4) | 10 (3.1) | 7 (2.2) | 11 (3.4) |
| | 50 and above | 18 (5.6) | 37 (11.6) | 4 (1.2) | 6 (1.9) | 7 (2.2) |
| **Residence** | Hanoi/HCMC | 93 (29.1) | 135 (42.2) | 38 (11.9) | 17 (5.3) | 39 (12.2) |
| | Other cities/ provinces | 65 (20.3) | 101 (31.6) | 19 (5.9) | 13 (4.1) | 23 (7.2) |
| **Education** | Secondary school | 9 (2.8) | 32 (10.0) | 3 (0.9) | 4 (1.2) | 3 (0.9) |
| | High school | 46 (14.4) | 67 (20.9) | 14 (4.4) | 9 (2.8) | 15 (4.7) |
| | Undergraduate/ Postgraduate | 103 (32.2) | 137 (42.8) | 40 (12.5) | 17 (5.3) | 44 (13.8) |
| **Religion** | None | 121 (37.8) | 165 (51.6) | 47 (14.7) | 18 (5.6) | 50 (15.6) |
| | Buddhism | 31 (9.7) | 58 (18.1) | 8 (2.5) | 11 (3.4) | 11 (3.4) |
| | Christianity | 6 (1.9) | 13 (4.1) | 2 (0.6) | 1 (0.3) | 1 (0.3) |

### Awareness and attitude in relation to MIA and comparison to CDA

Among nearly 400 respondents, only 87 participants, approximately 22%, had heard of MIA before participating in the survey, while most (77.9%) were not previously aware of MIA. After a brief description of MIA was provided, we asked for the initial perception of the respondents on the method. Most of the participants (90.1%) considered MIA an effective method to establish accurate CoD, however, only 68.8% would like to know more about MIA. There was no difference between the HCW and the community groups in terms of the awareness of MIA as well as knowing MIA as a method to determine CoD (Table 4).

We also explored the preference regarding either CDA or MIA. Among the respondents, the proportion of people who would agree to give consent to MIA was similar to that of CDA (83.24%, 82.74%, respectively). However, the number of people who were not sure of their decision to perform MIA was slightly higher than that for CDA (13.71%, 11.93%, respectively). Between the HCW and the community groups, the proportions of the HCWs who would give consent to either CDA or MIA were slightly higher than those in the community group (86.5% versus 81.9% for CDA and 87.8% versus 82.2% for MIA). Preference to either CDA or MIA across different socio-demographic groups within the community group was also examined (Table 5). A higher number of participants who were above 40 would consent to MIA rather than CDA, while the proportion of respondents under 40 who would give consent to MIA were slightly lower. More participants practicing Buddhism would prefer MIA, while a higher proportion of Christians would give approval for CDA to be conducted. In the education subgroups, the percentages were relatively similar.

The information in Table 6 showed that more than a third of the participants thought that MIA might be more acceptable among those who opposed CDA, while a considerable percentage (nearly 45%) of people were not certain whether MIA would be preferred to CDA. The reasons for preferring MIA were also explored, in which the minimally invasive nature of the method was regarded as its strength by nearly 90% of the participants. This reason aligned with the reason for disapproving CDA as mentioned above that the participants were worried about CDA causing considerable invasions on the deceased's body.

## Discussion

In this study, we explored the awareness and perceptions of both community members and HCWs regarding methods for determining CoD (i.e. CDA, MIA). In summary, most respondents perceived the benefits of CDA to determine CoD and detect hereditary diseases for early treatment. CDA was considered important for both medicine and forensic medicine specifically, and neccessary to conduct in the case of questionable deaths. On the contrary, those who opposed CDA were mainly concerned about the invasive nature of the method. The MIA method was not well known among respondents. However, a majority of participants agreed that MIA could help establish reliable CoD, after being explained about the method. Considering the acceptability levels between the two methods, we found that many respondents did not equate opposing CDA with consenting to MIA even though some participants thought MIA might be more acceptable because of its minimally invasive nature as well as convenience.

**Table 4. Awareness of MIA between HCWs and community members.**

| Awareness of MIA | HCWs ($n_1$ = 74) | Community ($n_2$ = 320) | p |
|---|---|---|---|
| | n (%) | n (%) | |
| **Having heard of MIA** | 17 (23.0) | 70 (21.9) | 0.84 |
| **Awareness of MIA as a method to determine CoD** | 68 (91.9) | 287 (89.7) | 0.57 |

**Table 5. Consent to either CDA or MIA in the community among socio-demographical groups (N = 320).**

| | | Consent to CDA | Consent to MIA |
|---|---|---|---|
| | | n (%) | n (%) |
| **Age group** | Under 30 | 77 (24.1) | 73 (22.8) |
| | 30–39 | 77 (26.6) | 69 (24.1) |
| | 40–49 | 63 (19.7) | 68 (21.2) |
| | 50 and above | 37 (11.6) | 45 (14.1) |
| **Residence** | Hanoi/HCMC | 136 (42.5) | 139 (43.4) |
| | Other cities/ provinces | 126 (39.4) | 124 (38.8) |
| **Education** | Secondary school | 30 (9.4) | 32 (10.0) |
| | High school | 83 (25.9) | 81 (25.3) |
| | Undergraduate/ Postgraduate | 149 (46.6) | 150 (46.9) |
| **Religion** | None | 190 (59.4) | 188 (58.8) |
| | Buddhism | 58 (18.1) | 64 (20.0) |
| | Christianity | 14 (4.4) | 11 (3.4) |

First and foremost, it is worth highlighting that due to the sampling methods, the study sample might not be representative, therefore, the results presented here might not be able to reflect the general perceptions of the whole Vietnamese population. The percentages of the participants who were younger than 40 years old and achieved at least university education were higher than national estimates. More than 60% of the participants were young, under 40, while only around 30% of the population were adults under 40 years old [15]. Further, the majority of the respondents had at least a university degree, whereas in the general population only 58.1% of Vietnamese people finished high school [16]. More than half of the participants included in this study lived in the two biggest cities of the country while according to the national statistics, in 2021, only around 18% of the population lived in these cities [15]. The proportion of participants who did not practice any religion was relatively similar to the national statistics of 73.6% [17].

Despite the fact that the population included in our sample is not completely representative of Vietnam, the results showed several commonalities found in previous studies on similar topics. Recent research from Nepal and Pakistan with the general public showed similar findings to our study about the benefit of CDA in identifying CoD, especially in cases of (forensically) suspicious death [18, 19]. They also found similar reasons for objecting autopsy. Particularly, in this study, body disfigurement from autopsy served as one of the main reasons for disapproving the method. This was also frequently found in the context of studies conducted in other LMICs, such as Nepal and Pakistan with participants from the general public [18, 19]. While participants from our study largely identified as non-religious (i.e. 75% of participants), their reasons expanded to more traditional or customary beliefs surrounding the body and death. [18, 19]

**Table 6. The attitudes of the participants towards CDA versus MIA.**

| | | Frequency | Percentage (%) |
|---|---|---|---|
| **People who oppose against CDA might give consent to MIA** | Yes | 141 | 35.8 |
| | No | 76 | 19.3 |
| | Unsure | 177 | 44.9 |
| **Reasons for giving consent to MIA instead of CDA (n = 36)** | Because MIA does not deform the dead body | 31 | 86.1 |
| | Because MIA is easier to conduct and more time-saving | 24 | 66.7 |
| | Because MIA is more acceptable in your religion | 5 | 13.9 |

In 2018, HCWs from Mozambique, including both formal and informal settings such as traditional healer facilities, reported different concerns about CDA. These ranged from the difficulty in obtaining informed consent from family, to the significant physical and mental burden on the HCWs conducting it, and to the risk of infection and stigma experienced among HCWs who performed the CDA [20]. The benefits of CDA, viewed by HCWs in Mozambique was to improve future diagnosis, which was contradictory to what our respondents who reflected more on CDA's role in indiviual benefits, such as the detection of hereditary disease.

Studies on the acceptability of CDA were also conducted with specific groups including bereaved families. In South Africa, a study focused on bereaved mothers was conducted in 2017 and showed similar findings with our study. Most bereaved mothers provided consent for autopsy on their stillborn children in order to know the CoD, which they believed would help them to better plan and manage their own pregnancies in the future [21]. Among this specific group and in the context of autopsy on stillborn babies, the reasons for disapproving CDA were found to be similar with our study from the general public including the concerns about the invasive nature of the procedure to the body and going against religious beliefs [21].

Religious factors, however, are not always a barrier to the implementation of CDA. Research conducted in Tanzania with multiple participant groups demonstrated that autopsy was accepted and religious factors were facilitators to the acceptance, not barriers. Conducting CDA in this context was often done to learn the cause of death thus being able to determine if witchcraft was involved, which were sourced from local religious beliefs, especially in case of sudden death of a person who had no illness history [22].

Regarding MIA, findings from many studies agreed with our findings that it could be useful to identify CoD. For studies among the general population, and for participants who had not heard of the method but had the procedure explained, they believed it could help to establish accurate CoD [20, 23, 24]. The minimally invasive nature of MIA was also considered a factor for people to prefer MIA over CDA, especially in studies of bereaved relatives. Bereaved respondents from a study in Rwanda agreed that one of the advantages of MIA was that it did not deform or alter the body while still being able to determine CoD [23]. Expanding upon our survey's finding, this study also added that CoD identifed by MIA can prevent family conflict and dispute among community members.

For studies including HCWs, participants had detailed concerns about MIA as well as mentioned several barriers to conducting MIA, even though the majority had never conducted or seen MIA [20]. HCW participants from the study in Southern Mozambique agreed that MIA would be easier, more convenient and time-saving to conduct, as well as less discomforting and would carry a lower chance of infection while performing compared with CDA [20]. However, some participants were also concerned about the method's reliability and potential delays on burial plans. There are also a large number of "invisibility at death" in which the deceased pass away at home, are buried secretly, thus keeping the healthcare staff from knowing about the death [20]. In contexts where death and burials occur outside the formal health system, the opportunity to discuss CDA or MIA with families may be absent.

From our findings, we realized that people who oppposed CDA, might not give consent to MIA, stating that they were unsure whether or not they would consent for MIA, however, they agreed MIA could be more acceptable on religious grounds. This finding shares a similar viewpoint with a study from UK with Muslim and Jewish community members who shared that MIA could only be approved if there were "multiple unexplained losses" and the option of non-invasive autopsy was not available [24]. Especially among Muslim participants, MIA was not really acceptable because it still contradicted their religious beliefs, as a participant responded "while it's still forbidden, it's less forbidden (compared to full autopsy)" [24]. This finding from our survey will be explored in more detail in the qualitative component.

There are a number of limitations to this work. First, we have not yet collected ethnographic data to explore the concerns and factors that may impact acceptance to perform or consent MIA in more detail, or the specifics of the logistics that might make MIA more acceptable in Vietnamese contexts. This will be covered in the subsequent qualitative component. Second, as discussed earlier, the majority of the respondents were under 40, lived in urban areas or obtained university education, which might be due to the fact that the data collection was limited to those who either worked or visited the NHTD or found the link on the website despite our efforts to circulate it to different groups among our networks. There may be different perceptions about death, dying, and the body outside of this group, for example, in ethnic minority groups where the death rituals are different than in Hanoi or in an older respondent group. Finally, although we had made the questionnaire completely anonymized from the start, the 80% of participants saying that they would consent to either CDA or MIA is potentially biased by the sampling method through the healthcare system when the respondent had to fill in the survey with HCWs in the room.

## Recommendations

Future studies on similar topics should explore further the socio-cultural barriers against CDA and MIA across different groups with representative samples that would enable comparisons between groups and between the sample and the whole population. The concerns of participants described in this study are also helpful for the development of public engagement materials to increase the awareness and acceptability of MIA among the general public.

## Conclusions

In conclusion, while postmortem sampling is not common in Vietnam, most participants were aware and in favor of CDA, especially when CoD was unclear. Its perceived benefits were mostly related to individual benefits, i.e. to detect diseases within the family for early treatment, not benefits to science and treatment improvements. The majority of the participants had not heard of MIA, but when explained, 35% of participants thought it would be more acceptable (while 45% did not know). The findings here provide an important starting point for more in-depth research among different groups using qualitative methods.

## Supporting information

**S1 Data.**
(SAV)

## Acknowledgments

We extend our gratitude to all the participants who took the time to participate in this study. We would also like to acknowledge our collaborators at the National Hospital for Tropical Diseases in Hanoi, Vietnam and the funders of the study.

## Author Contributions

**Conceptualization:** Ngan Ta Thi Dieu, Rogier van Doorn.

**Formal analysis:** Ngan Ta Thi Dieu, Nhung Doan Phuong, My Nguyen Le Thao, Mary Chambers, Rogier van Doorn, Jennifer Ilo Van Nuil.

**Writing – original draft:** Nhung Doan Phuong, My Nguyen Le Thao, Rogier van Doorn, Jennifer Ilo Van Nuil.

**Writing – review & editing:** Ngan Ta Thi Dieu, Mary Chambers, Duy Manh Nguyen, Ha Thi Lien Nguyen, Huong Thi Thu Vu, Thach Ngoc Pham.

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
