## [Decision Letter · Decision Letter 0]

8 Jan 2023

PGPH-D-22-01634

Knowledge and attitudes toward complete diagnostic autopsy and minimally invasive autopsy: a cross-sectional survey in Hanoi, Vietnam

Dear Dr. Van Nuil,

Thank you for submitting your manuscript to PLOS Global Public Health. After careful consideration, we feel that it has merit but does not fully meet PLOS Global Public Health’s publication criteria as it currently stands. Therefore, we invite you to submit a revised version of the manuscript that addresses the points raised during the review process.

We look forward to receiving your revised manuscript.

Kind regards,

Sadia Shakoor

Academic Editor

Journal Requirements:

a. Please clarify all sources of funding (financial or material support) for your study. List the grants (with grant number) or organizations (with url) that supported your study, including funding received from your institution. 

b. State the initials, alongside each funding source, of each author to receive each grant.

2. Please provide separate figure files in .tif or .eps format.

Additional Editor Comments (if provided):

Reviewers' comments:

Reviewer's Responses to Questions

**Comments to the Author**

1. Does this manuscript meet PLOS Global Public Health’s publication criteria? Is the manuscript technically sound, and do the data support the conclusions? The manuscript must describe methodologically and ethically rigorous research with conclusions that are appropriately drawn based on the data presented.

Reviewer #1: Partly

Reviewer #2: Partly

2. Has the statistical analysis been performed appropriately and rigorously?

Reviewer #1: Yes

Reviewer #2: Yes

3. Have the authors made all data underlying the findings in their manuscript fully available (please refer to the Data Availability Statement at the start of the manuscript PDF file)?

Reviewer #1: Yes

Reviewer #2: Yes

4. Is the manuscript presented in an intelligible fashion and written in standard English?

Reviewer #1: Yes

Reviewer #2: No

5. Review Comments to the Author

Reviewer #1: The presented research article is well structured and written, and addresses the challenges of CoD surveillance in LMICs, taking into account peoples’ beliefs and constraints towards CDA or MIA. The research question and its importance for further use are clearly expressed. The context of the study and its study population as well as the methods and materials used are adequately reported, even though the chosen approach might not really answer the stated research question or problem. It is expressed that the compliance of people to agree in CDA/MDA is particularly low in certain provinces and due to certain religious beliefs and cultural traditions. Nonetheless, there are mainly young, presumably healthy people of mainly 2 cities and with mainly high education within the study population, certainly due to the chosen method of acquisition which was prone to not reach vast groups of the population. These pitfalls of the study are discussed later on, but as they are very obvious to cause major biases the authors should consider to put them more prominent in the discussion section, before comparing their results to other studies of “the general public”. Otherwise, the discussion is concise and addresses the importance of the subject in many different LCMIs/ cultural contexts.

In the following, I have some suggestions that the authors might want to consider:

Abstract

reads in some sentences a little clumsy – please carefully improve language if possible

M&M

- The sentence “Since we aimed to obtain initial understandings of what people know and think of CDA and MIA, instead of examining the awareness and attitudes of the general public, no formal sample size calculation was done.” is not clear to me.

Results

- Socio-demographical characteristics: it would be really interesting to know – in comparison – how representative this is regarding the population in Vietnam in terms of age groups, religion and education. Maybe something could be added in this direction? This would help to be able to follow in the discussion later on why these results are only of limited use.

- In general, the authors should consider to analyse all their findings comparatively for the different groups. It might be good to split the data into HCW and non-HCW, as it is later, in the discussion, established that HCW often have reservations regarding CDA/MIA because they don’t feel comfortable performing it or even are stigmatized for performing it. This makes these two groups difficult to compare. For the other group that the authors call community the differences among the age groups, education level, site of residents would be highly interesting. Even if data is not enough for statistical testing, it could simply be descriptive (tables).

- Reasons for opposing against CDA: It would be interesting to know the distribution of the reasons amongst the different groups.

- Preference regarding consent to either CDA or to MIA: This seems to be one of the most crucial questions! Should be elaborated more on, also on the differences observed in the different strata of the surveyed sample population (e.g. age, residence, HCW).

Discussion

- It would be better to first discuss the (non-)representativeness of the surveyed sample population for the general public, before comparing the results with other studies and arguing about observed similarities (because the found distributions might be very biased).

Reviewer #2: Dear Sir/Mad.,

REVIEWER COMMENTS ON: “KNOWLEDGE AND ATTITUDES TOWARD COMPLETE DIAGNOSTIC AUTOPSY AND MINIMALLY INVASIVE AUTOPSY: A CROSS-SECTIONAL SURVEY IN HANOI, VIETNAM”.

The research topic is a good one and the authors’ zeal to help find solution to an identified gap in medical diagnosis is highly recommendable. It is a study that would add to knowledge acquisition. However, the authors have made many typographical errors that would affect the beauty of this important study. Some recommendations are given in continues prose form under each heading, which when modified would help to improve the quality of the paper for publication.

Abstract

In the background of the abstract, insert “the” between the “Knowing” and “cause” of …

Under “Methods/Results” the following editing should be done: Insert “used” in between “study” and “was” so that it becomes: The “study used was cross-sectional” … The authors did not a single questionnaire. So for the same sentence first sentence, they should remove the “a”, make the “questionnaire” plural and also change “was” to “were” and also replace “in” after hospital with a comma “,”. That sentence could now become: “The study used was cross-sectional, using semi-structured questionnaires that were disseminated via several websites and as paper-based in a national level hospital, Vietnam.”

The second sentence should be re-phrased. Then, “were” should be removed from the third (3rd) sentence: “We included 394 surveys were in the analysis.” The words “an”, “issue” and “for” in the last sentence under the conclusion part of the abstract should be edited by removing “an”, changing issue” to “issues” and “for” to “against”.

Introduction

The phrase “Knowledge of cause of death” in the first (1st) sentence under the introduction should be modified by adding “the” between the “of” and “cause”. For the first (1st) sentence of paragraph two (2), a full stop should be placed after “specimens”. After that the first letter that follows it should start another sentence by writing it in capital “For”. Place a comma “,” after “encephalitis” and “particular” then a semi colon after the second specimen before “whereas”.

The second sentence: “The reasons for this are threefold: most deaths occur outside the health system (patients are often discharged to die at home when prognosis is unfavorable), there are no facilities or trained personnel for full autopsy and subsequent pathological and microbiological examination in most provincial and referral hospitals, and a final major factor limiting consent to autopsy are negative attitudes toward post-morten examination, due to religious beliefs and cultural traditions.” in the fourth (4th) paragraph is too long. It should be re-phrased.

The fourth (4th) word “this” and the sixth (6th) word “threefold” should be changed into plural with a full stop placed after “unfavorable” in the bracket. “Secondly”, should be placed between the first word that follows “unfavorable”: “there”, with a full stop “.” replacing “comma” after “hospitals”. Then, “lastly” or “finally” should replace “and” with a comma, then the remaining words that end at “traditions”. References should be provided after it. Starting from “Minimally invasive autopsy …”, move them down to start a new paragraph. Full stop should be after “CDA”. Then “recent” should be started as “recently” with a comma “,” and a full stop “.” after the reference indicate “(6)”. The word “and” should be removed. Then, the first letter of the next word “the” should start with as capital. It should start as: “The Pneumonia Etiology Research …”

Starting from “Both studies …” to “open autopsy”, there should be a full stop after “autopsy” with the first letter of “using” started as capital. The first sentence of paragraph five (5) which would be paragraph six after the fourth paragraph is divided into two (2); should be divided into two sentences. In the seventh (7th) paragraph (or eighth, 8th) if paragraph is divided, the in-text reference: “Le Van Loi and Hoang Thi Lan (2019)” should be sited using the format for in-text referencing. Rephrase sentences that are too long especially those that ere highlight as yellow.

Autopsy is known to be associated with so many negative perceptions. Though demystifying people’s perception about autopsy is not one of the authors’ specific objectives, it would be advisable to tell readers what autopsy is.

Materials and Methods

Use the same recommendation given in the abstract for the questionnaires under this headline.

Results and Discussions

A Table 1 of “Soci-demographical characteristics of the participants” is given. The authors have only listed or indicated the various demographical characteristics but they did not use any of them as example to let readers understand why the table is necessary. Authors should briefly explain in at least two or three sentences how the demographic characteristics of the respondents influence their knowledge and attitude about CMA and MIA. They should use some of the demographic attributes of the respondents as examples in their explanations.

The authors also stated that “394 surveys were included in the analysis” but this figure cannot be identified from table 1. To solve this issue a row each should added to each group/variable to take care total number of participants in each group. Per their explanation, it means that “394” respondents were used for the survey. So, the total of each group or variable like “Age, City/Province, Education, Occupation” or “Religion” should be “394”. The total for “City/Province, Education”, and “Religion” were each “394” which is correct. But for the two variables, the total for each exceeded “394”. If there is/are any explanation(s) for these, they should be provided.

The next issue with table 1 too is that, they have tried to categorize the respondents, which is a good idea. But their first category “18-30” is not of the same intervals like the rests. Authors should re-categorize their respondents to ensure that they have equal intervals or they can use “below 30” to avoid the issue of unequal intervals.

Just like in Table 1, a row each should be added each variable/group for Tables 2-6 too. Then, the total percentages for each variable/group must be hundred (%). If there is/are any reason(s) why the percentages for variable must exceed (%), then they must be made known to the readers. In the same way, the total for each variable is supposed to be “394”. If there is/are any reason(s) why the total must exceed this number; then readers must know about that.

The same issues raised about the tables are applicable to the figures (Fig. 1, and Fig. 2.). Authors should ensure that the total of each variable, like respondents for the various categories under “Age” or “Religion” is “394”. If the total exceeds “394”, clear explanation(s) should be given to the readers.

The authors have done well with discussions that provided them with clues for suggestions or recommendations. However, there are still some modifications that need to be done to make their more standardized and understanding. The second sentence of paragraph one (1), under the discussion should be re-phrased to help make it more meaningful. For the first paragraph, authors should compare their findings to what literature says and cite the source of literature. The first two words in the third (3rd) sentence of paragraph two (2) should be re-phrased. Example, it could be “Some participants” or “For some participants”/respondents from our study …

Conclusion, Recommendations or Suggestions

Though the authors have done well by giving a very good conclusion, one of the major reasons why research is done is the ability to give suggestions or recommendations based on the findings to help solve a particular problem. Their results and discussions have provided many clues that they can be used for suggestions or recommendations. Example, if “The MIA method was not well known among respondents”, what is the likely suggestion or recommendation? Or, if amongst MIA and CDA; your respondents are saying that the “MIA might more acceptable”, which of the two would be recommended for stakeholders?

The authors add a sub-heading: “Suggestions/Recommendations” after they have concluded. Then, they should provide suggestions or recommendations to help stake holders to decide what they can do to solve the problem.

The manuscript should be accepted after the identified corrections are edited accordingly. The areas of the manuscript that need editing are highlighted with a gold/yellow Color.

6. PLOS authors have the option to publish the peer review history of their article (what does this mean?). If published, this will include your full peer review and any attached files.

**Do you want your identity to be public for this peer review?** For information about this choice, including consent withdrawal, please see our Privacy Policy.

Reviewer #1: No

Reviewer #2: No

---

## [Editor Report · Decision Letter 1]

23 Feb 2023

Knowledge and attitudes toward complete diagnostic autopsy and minimally invasive autopsy: a cross-sectional survey in Hanoi, Vietnam

PGPH-D-22-01634R1

Dear Dr. Van Nuil,

We are pleased to inform you that your manuscript 'Knowledge and attitudes toward complete diagnostic autopsy and minimally invasive autopsy: a cross-sectional survey in Hanoi, Vietnam' has been provisionally accepted for publication in PLOS Global Public Health.

Best regards,

Sadia Shakoor

Academic Editor
